# Cooperative Anti-Deception Jamming in a Distributed Multiple-Radar System under Registration Errors

**DOI:** 10.3390/s22197216

**Published:** 2022-09-23

**Authors:** Shanshan Zhao, Minju Yi, Ziwei Liu

**Affiliations:** 1College of Electronic and Optical Engineering, Nanjing University of Posts and Telecommunications, Nanjing 210023, China; 2College of Communications and Information Engineering, Nanjing University of Posts and Telecommunications, Nanjing 210023, China

**Keywords:** distributed multiple-radar system, registration error, deception jamming, anti-jamming, correlation coefficient, false target discrimination

## Abstract

A distributed multiple-radar system has natural advantages in anti-deception jamming. However, most of the anti-jamming methods are proposed in full spatial registration. In practice, the registration error is difficult to eliminate, which will seriously degrade the performance of cooperative anti-jamming. Therefore, it is of great significance to consider the problem of cooperative anti-deception jamming under registration error. In this paper, the cooperative anti-deception jamming method is proposed in a distributed multiple-radar system under registration errors. On the premise of the known registration error, target received signal vectors are estimated from an uncertainty region in each channel by maximum likelihood (ML) algorithm. With the estimated received signal vectors, a target discrimination algorithm is introduced based on the difference in target spatial scattering characteristics, which calculate the correlation coefficient between different target received signal vectors and discriminate a false target with a designed threshold. Furthermore, since the registration error depends on the radar site errors, theoretical derivation for the registration error is given as a function of the transmitter and receiver site errors. Finally, simulation results verify the feasibility of the proposed discrimination method, and its performance due to the influence of the jamming-to-noise ratio (JNR), the registration error, the target size, and the discrimination threshold are considered.

## 1. Introduction

Electronic warfare has become an important part of modern warfare [1,2]. As critical components of electronic warfare, electronic countermeasure (ECM) and electronic counter-countermeasure (ECCM) compete with and promote each other [3,4]. Deception jamming, as an effective category of ECM, has attracted more attention because of its high efficiency. With the development of digital radio frequency memory (DRFM) [5], the deception jammer can generate large amounts of active false targets simultaneously to mask the desired targets by modulating and retransmitting the intercepted the radar signals. The deception ability has been greatly enhanced, and the generated false targets will saturate the target extraction and tracking algorithms. Therefore, the ECCM ability of radar systems is of great importance for the survival and operation performance of electronic warfare. 

Almost all modern radars have implemented ECCM strategies. Monostatic radar can exploit pulse diversity [6,7,8], polarization character [9], motion feature [10,11,12], and DRFM quantization error [13,14,15] to counter deception jamming. However, the anti-jamming ability of monostatic radar is limited with its single view angle and deficient information obtained. Compared with the monostatic radar, a distributed multiple-radar system, consisting of transmitters, receivers, and a joint processing center [16,17], radiates the target from sufficiently different directions, and the obtained information can be processed by the joint processing center. Due to its widely separated transmit/receive antennas, a distributed multiple-radar system has natural advantages in terms of anti-deception jamming. Available anti-deceptive jamming methods in multiple-radar can be divided into data-level fusion algorithms [18,19,20,21,22] and signal-level fusion algorithms [23,24,25,26,27]. The data-level fusion algorithms jointly process the measurements (range, angle, and Doppler information) or the filtered tracks in each local radar to discriminate the false target, based on the fact that deception jammers can hardly generate false target signals with matched range and velocity relationship to each station. With the improvement in synchronization technology and communication capabilities, signal-level fusion algorithms, instead of data-level fusion algorithms have inevitably become the major development tendency. The signal-level fusion algorithms counter the deception jamming by fusing the received signals from the local radar sites with less information loss compared with data levels, which explores more information and therefore can achieve better anti-jamming performance. On account of a target’s RCS spatial variations [28,29,30], the target echoes in distributed stations are decorrelated, while the deception signals are fully coherent. This difference in target spatial scattering characteristics serves as the theoretical basis of signal-level fusion anti-jamming algorithms. However, all of the available signal-level fusion methods are designed under the assumption of full spatial registration. 

Actually, full spatial registration is almost impossible due to the existence of systematic errors, measurement errors, site errors, and time errors [31,32]. In particular, the airborne radar and other moving platforms are in the scene of fast maneuvering or maneuvering turning. Due to the deviation of the satellite navigation system, each radar will also bring positioning deviation, which increases the difficulty of spatial registration. To eliminate the non-random static registration error, a large number of registration algorithms have been proposed, which can be categorized as measurement-level registration algorithms [33] and track-level registration algorithms [34,35,36]. The measurement-level registration algorithms establish the model by associating the registration error with the measurement data, and maximum likelihood (ML) algorithm or least square (LS) algorithm are used to reduce or eliminate the registration error. In the process of tracking fusion, Kalman algorithm or probability hypothesis density algorithm are applied to estimate and eliminate the registration error for the track-level registration algorithm. When faced with the random registration errors, these methods are not effective. Therefore, some registration errors will inevitably remain after error calibration, which should be considered in practice.

In presence of the registration error, a data-level fusion anti-deception jamming algorithm is proposed in [37], considering the effect of radar site errors in the target discrimination model. However, so far, there has been no corresponding discussion of signal-level fusion anti-jamming methods for distributed multiple-radar. Only in the application of cooperative target detection, has the effect of registration error on signal-level fusion algorithms been considered [38]. Sliding window method is applied to select limited observation data in each spatial diversity channel, which has certain reference significance for this paper.

It motivates us to develop the cooperative anti-deception jamming methods in distributed multiple-radar systems under registration errors. With the registration error known, the target discrimination method is designed, which can be divided into two steps. In the first step, an ML algorithm is applied to estimate target received signal vectors in each channel from an uncertainty region, the size of which is determined by the registration error. In the second step, a target discrimination algorithm is proposed based on the correlation coefficient between difference estimated target received signal vectors. Then, the relationship of the registration error with the radar site errors is derived theoretically and simulated. Finally, numerical simulation analysis verifies the effectiveness of the proposed method.

The remainder of the paper is organized as follows. Section 2 describes the problem of spatial registration errors and introduces the signal model of target discrimination in a distributed multiple-radar system. Section 3 proposes the target discrimination method, including the target discrimination with the registration error known and the derivation of the registration error. In Section 4, numerical simulation analysis is provided. Finally, conclusions are drawn in Section 5.

## 2. Problem Description and Signal Model

A distributed multiple-radar system with *M* transmitters and *N* receivers is considered to detect enemy aerial targets. In the aerial surveillance area, there are multiple targets and independent jammers. The jammer can generate multiple false targets to protect the true targets. In the distributed multiple-radar system, *MN* transmitter–receiver channels, named spatial diversity channels, can be formed. 

The detection area is divided according to the space resolution cell (SRC), which is the intersection of resolution cells of each pair of the transmitting and receiving stations. For each SRC, the received signal in the *MN* transmitter–receiver channels are collected to form a received signal vector, which is used for the subsequent target detection and false target discrimination. 

The space division model for distributed radar with two spatial diversity channels is shown in Figure 1, where Figure 1a gives the case of full registration and Figure 1b gives the case with registration error. For the convenience of derivation, a two-dimensional plane model is used here. The mesh split by parallel lines is the SRC, whose edge length is about the size of the radar range resolution, and the error ellipse represents the range-angle resolution cell (RARC) for each spatial diversity channel. Due to the difference in range resolution and angular resolution, the RARC of each spatial diversity channel covers multiple SRCs. The black dot indicates the location of the under-test target in space.

In the absence of registration error, as shown in Figure 1a, the RARC to be detected in the two spatial diversity channels exactly matches the target position. The target received signal vector is obtained by sampling the echo signals in each space diversity channel in RARC. However, the motion characteristics of the moving platform make it difficult for each radar to achieve full registration. As shown in Figure 1b with registration error, the RARC in two spatial diversity channels cannot match the target position, and the sampling of the target echo may be missed in the received signal vector. 

Due to the registration errors, the RARC containing the target echo signal in each spatial diversity channel will appear in an uncertainty region near the detected area. The uncertainty region is the expansion of the fusion dot in the range and azimuth, as shown in Figure 2. The greater the registration error of the space diversity channel, the greater the number of RARCs contained in the uncertainty region. To obtain the sampling of the target echo, multiple RARC samples of the uncertainty region in each spatial diversity channel are used to construct the received signal vector. Therefore, the target received signal vector is contained in the limited set of observations obtained in the spatial diversity channel.

Suppose that the size of registration error is *Q* RARCs that is determined by the radar site error, an uncertainty region containing *Q* RARCs in the *mn*-th spatial diversity channel is denoted by $y_{mn}^{}=[y_{mn}^{}(1),y_{mn}^{}(2),\ldots ,y_{mn}^{}(Q)]$, m=1,2,…,M, n=1,2,…,N. Since the target exists only in one RARC, one sample is obtained from *Q* RARC samples in each channel to construct the received signal vector. It is assumed that $y_{mn}^{}(q_{mn}^{})$ refer to the target signal (true or false target). $q_{mn}$ is an unknown parameter, which should be determined before the construction of the target signal vector.

If one RARC sample of a channel contains a true target, the target echo signal sampling $y_{mn}^{}(q_{mn}^{})$ can be written as
(1)$$y_{mn}^{}(q_{mn}^{})=\alpha_{mn}\exp(-j2\pi R_{mTn}^{}/\lambda )+w_{mn}$$
where, $w_{mn}$ denotes the noise sample, which is an independent identically distributed complex Gaussian noise with variance of $\sigma_{mn}^{2}$ and zero mean, $w_{mn}\sim CN(0,\sigma_{mn}^{2})$; $\exp(-j2\pi R_{mTn}^{}/\lambda )$ is the carrier frequency residual. λ is the system wavelength, the range sum $R_{mTn}^{}=R_{mT}^{}+R_{Tn}^{}$, $R_{mT}^{}$ and $R_{Tn}^{}$ denote respectively the range from the *m*-th transmitter to the target and the range from the target to the *n*-th receiver.

According to the radar equation, the target amplitude $\alpha_{mn}$ in the *mn*-th channel is
(2)$$\alpha_{mn}=\lambda \sigma_{mn}\sqrt{P_{Tm}G_{Tm}G_{Rn}}/\left(4\pi \sqrt{4\pi }R_{mT}^{}R_{Tn}^{}\right)$$
where, $P_{Tm}$ is the transmitted power of the *m*-th transmitter; $G_{Tm}$ and $G_{Rn}$ are the antenna gain of the *m*-th transmitter and the *n*-th receiver; $\sigma_{mn}$ is the target radar cross section (RCS), which is modelled as a random variable following the complex Gaussian distribution, i.e., $\sigma_{mn}\sim CN(0,\varsigma_{mn}^{2})$.

If one RARC sample contains a false target generated by the deception jamming, the target signal sampling $y_{mn}^{}(q_{mn}^{})$ can be written as
(3)$$y_{mn}^{}(q_{mn}^{})=\beta_{mn}\exp(-j2\pi R_{mJn}^{}/\lambda )+w_{mn}$$
with $R_{mJn}^{}$ denoting the range along the path from the *m*-th transmitter to the jammer and then to the *n*-th receiver. The active false signal amplitude $\beta_{mn}^{}$
in the *mn*-th channel can be given as
(4)$$\beta_{mn}^{}=\upsilon \sqrt{P_{J}G_{Rn}}\lambda /\left(4\pi R_{Jn}\right)$$
where, υ denotes the possible amplitude fluctuations with unknown distribution; $P_{J}$ is the jamming power; $R_{Jn}$ is the range from the jammer to the *n*-th receiver.

For the other samples containing neither true nor false targets,
(5)$$y_{mn}^{}(q)=w_{mn},\;\;\;q=1,\ldots ,Q,\;q\neq q_{mn}$$

## 3. Target Discrimination Algorithm

On the premise that the registration error *Q* is known, a target discrimination algorithm is proposed based on the differences in the correlation coefficient between different target received signal vectors, which is estimated from the uncertainty region RARCs in each channel $y_{mn}^{}$. Since the registration error *Q*, depending on the radar site error, would affect the discrimination performance of the proposed algorithm, the registration error is then derived under different radar site errors.

### 3.1. Target Discrimination with Registration Error Known

The presence of registration errors will affect the correct acquisition of the target signal vector, which will seriously affect the performance of deception anti-jamming. To construct the target signal vector, the target signal in each channel should be estimated from the defined uncertainty region RARCs. With the estimated target received signal vectors, the target discrimination method is then proposed.

#### 3.1.1. Estimation of Target Received Signals

Target discrimination is applied to the detected targets in the target detection. For a detected target, there is a target echo signal $y_{mn}^{}(q_{mn}^{})$ in the uncertainty region RARCs $y_{mn}^{}=[y_{mn}^{}(1),\;y_{mn}^{}(2),\;\ldots ,\;y_{mn}^{}(Q)]$ for each diversity channel, and $q_{mn}^{}$ is unknown but determined. According the signal model in Section 2, for $q\neq q_{mn}$, $y_{mn}^{}(q)$ is modeled as an independent identically distributed Gaussian random variable with zero mean and variance of $\sigma_{mn}^{2}=1$. 

In the case of the detected target being a true target, $y_{mn}^{}(q_{mn}^{})$ follows the complex Gaussian distribution with zero mean and the variance of $\lambda_{i}$ (unknown). In the case of the detected target being a false target, the distribution of the received signal $y_{mn}^{}(q_{mn}^{})$ is determined by the jammer, which is assumed that follows the same distribution as the true target to obtain the best deception effect. Under the condition that the target locates at the $q^{\prime }$-th RARC, the joint conditional probability density function of the random vector $y_{mn}^{}$ can be written as
(6)$$\begin{array}{l} f(y_{mn}^{}|q^{\prime })\\ =\frac{1}{\pi (1+\lambda_{mn})}\exp\left(-\frac{y_{mn}(q^{\prime })y_{mn}^{*}(q^{\prime })}{1+\lambda_{mn}^{}}\right)\frac{1}{\pi^{Q-1}}\exp\left(-\sum_{q=0,q\neq q^{\prime }}^{Q-1}y_{mn}^{}(q)y_{mn}^{*}(q)\right)\\ =\frac{1}{\pi^{Q}(1+\lambda_{mn}^{})}\exp\left(-\frac{|y_{mn}^{}(q^{\prime }){|}^{2}}{1+\lambda_{mn}}-\sum_{q=0,q\neq q^{\prime }}^{Q-1}|y_{mn}(q){|}^{2}\right)\\ =\frac{1}{\pi^{Q}(1+\lambda_{mn})}\exp\left(\frac{\lambda_{mn}|y_{mn}(q^{\prime }){|}^{2}}{1+\lambda_{mn}}-\sum_{q=0}^{Q-1}|y_{mn}(q){|}^{2}\right) \end{array}$$
where, ${\left(\cdot \right)}^{*}$ denotes use of the complex conjugate. ML estimation is then used to determine the RARC where the target locates
(7)$$\begin{array}{l} q_{mn}=\underset{q^{\prime }}{\max}f(y_{mn}|q^{\prime })\\ =\underset{q^{\prime }}{\max}\left\{\frac{1}{\pi^{Q}(1+\lambda_{mn})}\exp\left(\frac{\lambda_{mn}|y_{mn}(q^{\prime }){|}^{2}}{1+\lambda_{mn}}-\sum_{q=0}^{Q-1}|y_{mn}(q){|}^{2}\right)\right\}\\ =\frac{1}{\pi^{Q}(1+\lambda_{mn})}\exp\left(\frac{\lambda_{mn}\underset{q^{\prime }}{\max}(|y_{mn}(q^{\prime }){|}^{2})}{1+\lambda_{mn}}-\sum_{q=0}^{Q-1}|y_{mn}(q){|}^{2}\right)\\ =\underset{q^{\prime }}{\max}(|y_{mn}(q^{\prime }){|}^{2}) \end{array}$$

Obviously, the sample with the largest energy in the RARC has the largest probability of being the sample of the target signal in each channel. This conclusion applies to true and false targets. 

Therefore, the received signal vector x can be obtained by using the maximum energy of the observations in each channel,
(8)$$x_{i}=[y_{11}(q_{11}),y_{12}(q_{12}),\ldots ,y_{MN}(q_{MN})]$$

#### 3.1.2. Target Discrimination Method

To discriminate active false targets generated by the deception jamming, the difference in spatial scattering characteristics between true and false targets is exploited. The target echoes in distributed stations are decorrelated due to the spatial diversity, while the deception signals from one jammer are highly correlated. This difference is reflected in the correlation coefficient between different target received signal vectors. For a radar target, the correlation coefficient with any target is small. For a false target, the correlation coefficient with the false targets produced by the same jammer is approximately one. Moreover, a jammer always generates a large number of false targets at a time to obtain better deception effect. Therefore, a discrimination rule can be developed: the target highly correlated with multiple targets is determined as a false target; otherwise, it is a radar target. 

It is assumed that *K* targets are detected in the detection area, and the received signal vector is $x_{k}$, k=1,2,…,K. The correlation coefficient between different targets is calculated, forming a correlation coefficient matrix Ω (K×K matrix). Its element of the *i*-th row and *j*-th column is
(9)$${\left[\Omega \right]}_{ij}=\frac{{\left(x_{i}\right)}^{H}x_{j}}{\left||x_{i}|\right|\left||x_{j}|\right|}$$
where, ${\left(\cdot \right)}^{H}$ denotes the conjugate transpose, and |⋅| denotes the Euclidean norm of a vector. 

In general, two targets can be considered as false targets when their correlation coefficient is greater than a certain value η. In the absence of any prior knowledge, η=0.5 may be reasonable, and it can also be adjusted in practical application to achieve better discrimination performance. After binary quantization, the quantified correlation coefficient matrix $\bar{\Omega }$ can be obtained,
(10)$${\left[\bar{\Omega }\right]}_{ij}=\left\{\begin{array}{r} 0,\;[\Omega {]}_{ij}<\eta \\ 1,\;[\Omega {]}_{ij}\geq \eta \end{array}\right.$$

The matrix $\bar{\Omega }$ is a symmetric matrix with diagonal elements all being a value of 1, and all elements are either 0 or 1. If ${\left[\bar{\Omega }\right]}_{ij}=0$, it is indicated that the received signal vectors of the *i*-th target and the *j*-th target are uncorrelated, these two targets are then recorded as the true target one time; if ${\left[\bar{\Omega }\right]}_{ij}=1$, it is indicated that the received signal vectors of the *i*-th target and the *j*-th target are correlated, these two targets are then recorded as the false target one time.

According to the quantization matrix $\bar{\Omega }$, the number of times when a target is recorded as a false target can be counted, forming an integer vector Φ,
(11)$${\left[\Phi \right]}_{i}=\sum_{i=1,i\neq j}^{K}{\left[\bar{\Omega }\right]}_{ij}$$

If the *i*-th target is a radar target, the correlation coefficient with any target is very small, ${\left[\Phi \right]}_{i}$ is a small integer. Especially in the independent case, no target is correlated with it, ${\left[\Phi \right]}_{i}$ is equal to zero. If the *i*-th target is a false target, the other false targets generated by the same jammer are all linearly related to it, ${\left[\Phi \right]}_{i}$ is a larger integer. 

To protect the target, a larger number of false targets is always generated once in the radar detection area. Therefore, we can discriminate the false targets according to the following criteria,
(12)$$\left\{\begin{array}{l} {\left[\Phi \right]}_{i}\leq \kappa ,\;\mathrm{the}\;i-\mathrm{th}\;\mathrm{target}\;\mathrm{is}\;a\;\mathrm{true}\;\mathrm{target}\\ {\left[\Phi \right]}_{i}>\kappa ,\;\mathrm{the}\;i-\mathrm{th}\;\mathrm{target}\;\mathrm{is}\;\mathrm{an}\;\mathrm{active}\;\mathrm{false}\;\mathrm{target} \end{array}\right.$$

In the ideal independent case, the discrimination threshold κ is zero. However, considering some low probability events, such as two true targets or a true target and a false target, happen to be highly correlated, κ can be set as a small integer. Generally, we can set κ = 1 or 2, and the simulation show that it can already obtain excepted discrimination performance.

The proposed method discriminates deception jamming based on the correlation coefficient between different received signal vectors without any prior information. It is obvious that it can be used in the scenario of multiple jammer sources. The received signal vectors of false targets generated by each jammer are approximately linearly related. Therefore, as long as the number of the false targets generated by a jammer exceed the threshold κ (κ is small, but the number of false targets is generally larger to obtain better deception performance, so this condition almost always holds), the proposed method can effectively discriminate the false targets generated by the jammer.

### 3.2. Derivation of the Registration Error

The target discrimination method is introduced with the known registration error, which is always unknown in practice. The registration error is caused by the site error of moving platforms. In this section, the derivation of registration error as a function of site errors is given. 

Using one spatial diversity channel as an example to discuss the relationship between the registration error and the site error. In this channel, the transmitter and the receiver locate at ${\left[x_{T},y_{T}\right]}^{T}$ and ${\left[x_{R},y_{R}\right]}^{T}$, where ${\left(\cdot \right)}^{T}$ denotes the matrix transpose. For the target, locating at $X={\left[x,y\right]}^{T}$, its measurement $Z={\left[\rho_{R},\theta_{R}\right]}^{T}$ in the receiver can be written as
(13)$$\left\{\begin{array}{l} \rho_{R}=\sqrt{{\left(x-x_{T}\right)}^{2}+{\left(y-y_{T}\right)}^{2}}+\sqrt{{\left(x-x_{R}\right)}^{2}+{\left(y-y_{R}\right)}^{2}}\\ \theta_{R}=\arctan\frac{y-y_{R}}{x-x_{R}} \end{array}\right.$$

To obtain the effect of the site error on the registration of target measurement, use differentiation on the both sides of (13),
(14)$$\left[\begin{array}{c} d\rho_{R}\\ d\theta_{R} \end{array}\right]=\left[\begin{array}{cccc} -c_{R1} & -c_{R2} & -c_{T1} & -c_{T2}\\ \frac{\sin\theta_{R}}{r_{R}} & -\frac{\cos\theta_{R}}{r_{R}} & 0 & 0 \end{array}\right]\left[\begin{array}{c} dx_{R}\\ dy_{R}\\ dx_{T}\\ dy_{T} \end{array}\right]$$
which can be rewritten as
(15)$$dZ=BdX_{s}$$
with the target registration error vector $dZ={\left[d\rho_{R},d\theta_{R}\right]}^{T}$ and the site error vector $dX_{s}={\left[dx_{R},dy_{R},dx_{T},dy_{T}\right]}^{T}$. Moreover,
(16)$$B=\left[\begin{array}{cccc} -c_{R1} & -c_{R2} & -c_{T1} & -c_{T2}\\ \frac{\sin\theta_{R}}{r_{R}} & -\frac{\cos\theta_{R}}{r_{R}} & 0 & 0 \end{array}\right]$$
(17)$$c_{l1}=\frac{x-x_{l}}{r_{l}}=\cos\theta_{l},c_{l2}=\frac{y-y_{l}}{r_{l}}=\sin\theta_{l}\;\;\;\left(l=R,T\right)$$

The terms $r_{R}$ and $r_{T}$ are the ranges from the target to the receiver and the transmitter, satisfying $\rho_{R}=r_{T}+r_{R}$. Based on (13), $r_{R}$ can be derived as
(18)$$r_{R}=\frac{\rho_{R}^{2}-\left[{\left(x_{R}-x_{T}\right)}^{2}+{\left(y_{R}-y_{T}\right)}^{2}\right]}{2\left[\rho_{R}+\left(x_{R}-x_{T}\right)\cos\theta_{R}+\left(y_{R}-y_{T}\right)\sin\theta_{R}\right]}$$

It is assumed that the site errors of the receiver and the transmitter follow the independent zero-mean Gaussian distribution with their standard deviations $\sigma_{x_{R}}=\sigma_{y_{R}}=\sigma_{s_{R}}$ and $\sigma_{x_{T}}=\sigma_{y_{T}}=\sigma_{s_{T}}$, respectively. Therefore, $\Delta X_{s}=X_{s}-{\bar{X}}_{s}\sim N(0,\Lambda )$, ${\bar{X}}_{s}$ denotes the actual location of transmitter and receiver. Then, the covariance matrix Λ of $\Delta X_{s}$ is
(19)$$\Lambda =E\left[dX_{s}dX_{s}^{T}\right]=\mathrm{diag}([\sigma_{x_{R}}^{2},\sigma_{y_{R}}^{2},\sigma_{x_{T}}^{2},\sigma_{y_{T}}^{2}])$$
where, diag(⋅) represents a diagonal matrix with the elements on its diagonal.

According to (15), the target registration error is a zero-mean Gaussian random variable, $\Delta Z=Z-\bar{Z}\sim N(0,P)$ and its covariance matrix P can be obtained,
(20)$$P=E\left[dZdZ_{T}\right]=B\Lambda B^{T}=\left[\begin{array}{cc} \sigma_{\rho_{R}}^{2} & 0\\ 0 & \sigma_{\theta_{R}}^{2} \end{array}\right]$$
where (21)$$\sigma_{\rho_{R}}^{2}=\left(c_{R1}^{2}+c_{R2}^{2}\right)\sigma_{s_{R}}^{2}+\left(c_{T1}^{2}+c_{T2}^{2}\right)\sigma_{s_{T}}^{2}$$
(22)$$\sigma_{\theta_{R}}^{2}=\left[{\left(\frac{\sin\theta_{R}}{r_{R}}\right)}^{2}+{\left(\frac{\cos\theta_{R}}{r_{R}}\right)}^{2}\right]\sigma_{s_{R}}^{2}$$

According to the covariance matrix of the registration error P, the error ellipse is an axis-aligned ellipse, that is, the major axis and the minor axis of the error ellipse are parallel to the range and azimuth dimension. The probability density function of target measurement $Z={\left[\rho_{R},\theta_{R}\right]}^{T}$ can be given as
(23)$$f(\Delta \rho_{R},\Delta \theta_{R})=\frac{1}{2\pi \sigma_{\rho_{R}}\sigma_{\theta_{R}}}\exp\left\{-\frac{1}{2}\left(\frac{{\left(\Delta \rho_{R}\right)}^{2}}{\sigma_{\rho_{R}}^{2}}+\frac{{\left(\Delta \theta_{R}\right)}^{2}}{\sigma_{\theta_{R}}^{2}}\right)\right\}$$

With k defining the size of the error ellipse, the registration error ellipse can be described as
(24)$$\frac{{\left(\Delta \rho_{R}\right)}^{2}}{\sigma_{\rho_{R}}^{2}}+\frac{{\left(\Delta \theta_{R}\right)}^{2}}{\sigma_{\theta_{R}}^{2}}=k^{2}$$

The target registration error locates in a certain uncertainty region S with probability α, then
(25)$$\alpha =\iint_{\Delta Z\in S}\frac{1}{2\pi \sigma_{\rho_{R}}\sigma_{\theta_{R}}}\exp\left\{-\frac{1}{2}\left(\frac{{\left(\Delta \rho_{R}\right)}^{2}}{\sigma_{\rho_{R}}^{2}}+\frac{{\left(\Delta \theta_{R}\right)}^{2}}{\sigma_{\theta_{R}}^{2}}\right)\right\}d(\Delta \rho_{R})d(\Delta \theta_{R})$$

According to [39], the size of the error ellipse k can be calculated,
(26)$$k=\sqrt{-2\ln(1-\alpha )}$$

Then, the two axes length of the error ellipse can be expressed as
(27)$$a=2\sqrt{-2\ln(1-\alpha )}\sigma_{\rho_{R}},\;b=2\sqrt{-2\ln(1-\alpha )}\sigma_{\theta_{R}}$$

In the error ellipse, axis a corresponds to the range dimension, and axis b corresponds to the angle dimension. The number of range resolution cells in the error ellipse is
(28)$$Q_{\rho }=\left\{\begin{array}{l} \left\lceil \frac{a}{\Delta R}\right\rceil ,\;\left\lceil \frac{a}{\Delta R}\right\rceil \;\mathrm{is}\;\mathrm{an}\;\mathrm{odd}\;\mathrm{number}\\ \left\lceil \frac{a}{\Delta R}\right\rceil +1,\;\left\lceil \frac{a}{\Delta R}\right\rceil \;\mathrm{is}\;\mathrm{an}\;\mathrm{even}\;\mathrm{number} \end{array}\right.$$

The number of angle resolution cells in the error ellipse is
(29)$$Q_{\theta }=\left\{\begin{array}{l} \left\lceil \frac{b}{\theta_{3}}\right\rceil ,\;\left\lceil \frac{b}{\theta_{3}}\right\rceil \;\mathrm{is}\;\mathrm{an}\;\mathrm{odd}\;\mathrm{number}\\ \left\lceil \frac{b}{\theta_{3}}\right\rceil +1,\;\left\lceil \frac{b}{\theta_{3}}\right\rceil \;\mathrm{is}\;\mathrm{an}\;\mathrm{even}\;\mathrm{number} \end{array}\right.$$
where $\left\lceil \cdot \right\rceil$ stands for rounding to the nearest integers greater than or equal to it, ΔR and $\theta_{3}$ refer to the range and angular resolution, respectively.

Therefore, the registration error Q, that is the number of RARCs in the error ellipse can be expressed as
(30)$$Q=Q_{\rho }Q_{\theta }$$

## 4. Numerical Simulation Analysis

The distributed multiple-radar system consists of M=2 transmitters and N=5 receivers to detect the area with an aircraft formation. Four aircrafts in the formation serve as radar true targets, two of which carry active jammers to perform deception jamming on the multiple-radar system. The locations of the transmitters, receivers, and aircraft targets under the Cartesian coordinate system are shown in Table 1.

It is assumed that the range resolution in each channel is 50 m, and the angular resolution is 2°. Use the channel consisting of the transmitter at [0, 0] m and the receiver at [500, 0] m, and the target at [30, 31] km as an example. The target registration error locating in a certain uncertainty region with probability 0.99 is calculated as a function of the site errors of transmitters and receivers according to (28), (29), and (30). Since the angular resolution is high and the target is far from the radar station, the number of angle resolution cells in the error ellipse $Q_{\theta }$ is always 1, i.e., there is no registration error in angle. The target registration error Q is equal to the number of range resolution cells in the error ellipse $Q_{\rho }$, and the simulated results are reported in Figure 3. It is obvious that the registration error is proportional to the site errors. In the case of the site error less than 50 m, the registration error is no larger than 9, and the uncertainty region contains up to 9 RARCs.

In the case of the registration errors consisting of 3, 5, and 7 RARCs, the proposed target discrimination method is simulated to evaluate its discrimination performance. Additionally, the existing method in [27] is simulated as a comparison with the registration error *Q* = 3. The antenna gain is assumed to be the same for all transmitter and receiver stations. The system wavelength is λ=0.1 m. To protect the aircraft formation, each jammer generates eight active false targets at a time. The SNRs of all true targets are the same, and are set to be 8 dB in the first channel. The JNRs for all active false targets are the same and change from 7.5 dB to 20 dB in the first channel. The SNR or JNR in other diversity channels can be calculated according to the radar equation. In the absence of any prior knowledge, the threshold η in (10) is set as η=0.5, and the target discrimination threshold κ in (12) is set as κ=2. Moreover, the discrimination performance is also determined by the correlation coefficient of true targets, which depends on the target location, target size *D*, and other factors [30]. Therefore, the discrimination performance varying with the target size *D* is given here. The target size *D* of all true targets is assumed all the same.

With 10^5^ Monte Carlo simulation trials, the discrimination probability of true targets $P_{T}$ and the misjudgment probability of active false targets $P_{F}$ varying with the JNR is simulated. The discrimination probability of the existing method [27] with *Q* = 3 and *D* = 30 m is shown in Figure 4. Obviously, the existing method can obtain satisfied discrimination probability of true targets; however, the false target misjudgment probability is always greater than 70% under different JNR. This is due to the fact that the noise samples brought by the registration error enhance the de-correlation of the real target and destroy the correlation of the active false target. The high misjudgment probability indicates that the existing method cannot discriminate active false targets under registration errors.

For the proposed method, its discrimination probability is reported in Figure 5, Figure 6 and Figure 7 in the case of three different registration errors *Q*. The curves in each figure correspond to different values of the target size *D*, where *D* = 0 m, 15 m, and 30 m.

As shown in Figure 5, Figure 6 and Figure 7, under all simulation conditions, the discrimination probability of true targets is always larger than 95%, and the misjudgment probability of false targets keeps less than 0.9%. In other words, the proposed method can obtain expected discrimination performance under registration error, discriminating the false targets effectively with the preserved true targets, which indicates the feasibility of the algorithm.

With the increase in the JNR, the misjudgment probability of false targets decreases until it is close to zero, as shown in Figure 5b, Figure 6b and Figure 7b. In the case of lower JNR, parts of the jamming received signal vector may be replaced by the random noise when ML algorithm is used to estimate the target received signals. As the JNR increases, the probability that the jamming signals in some channels are replaced by the noise decreases, which will enhance the high correlation of false targets generated by the same jammer. Besides, an increase in JNR would also lead to an increase in correlation, and the enhanced high correlation brings the decrease of false target misjudgment probability. It is shown in Figure 5a, Figure 6a and Figure 7a that there is an inflection point near 10 dB in the curves of the discrimination probability of true targets *P*_T_, the discrimination probability decreases with the JNR when JNR < 10 dB, and increases with the JNR when JNR > 10 dB. In the case of lower JNR, parts of the jamming received signal vector may be replaced by the random noise, increasing the randomness of the jamming signal vector. Then, the probability of random received signal vector of true targets related to that of false targets becomes lower, bringing the improvement on the discrimination performance for true targets. When JNR > 10 dB, the probability that parts of the received signals are replaced by the noise becomes extremely low, and the estimated received signal vector is close to the actual received signal vector. With the increase in the JNR in this case, the subspace of the received jamming signal vector becomes smaller, and the probability of a true target correlated with a false target becomes lower, caused the better discrimination performance for true targets. In practice, JNR is usually high to ensure that the detection threshold can be exceeded to generate active false targets, and the proposed method can achieve acceptable target discrimination performance.

Comparing the simulation results in Figure 5, Figure 6 and Figure 7, the effect of the registration errors on the discrimination performance is shown. It can be seen from Figure 5a, Figure 6a and Figure 7a that the increase in the registration error will lead to a slight improvement in the discrimination probability of true targets. As shown in Figure 5b, Figure 6b and Figure 7b, the registration error increases the misjudgment probability of active false targets when JNR < 10 dB. When the JNR is larger than 10 dB, the registration error has less effect on the misjudgment probability, which is almost zero. With the larger registration error, the corresponding larger uncertainty region makes it more likely that a portion of the target received signal vector will be replaced by the independent noise, which will enhance the independence of the true target and reduce the strong correlation of false targets. As a result, the discrimination probability of true targets and the misjudgment probability of active false targets are improved. 

Comparing the curves corresponding to different target size *D* in Figure 5, Figure 6 and Figure 7, the following conclusions can be obtained for the effect of target correlation coefficient on the method. On one hand, the proposed method can achieve higher discrimination probability of true targets with larger target size. The larger the target size, the lower the target correlation, which will increase the difference between the true target and the false target, leading to better discrimination performance for true targets. When the spatial scattering characteristics of true target in different channels are completely correlated with the target size being zero, the proposed discrimination algorithm can still obtain a discrimination probability of about 96%. This is due to the fact that the received signal vectors of different true targets are not highly correlated and caused by their different locations, and that the received signal vectors of true and false targets are also not highly correlated due to their different signal vector structures suffering from double-path or one-way attenuation. On the other hand, the target size has no effect on the misjudgment probability of active false targets, since the received signal vectors of the false targets generated by the same jammer are always highly correlated, which is independent of the target size.

In addition, the effect of the discrimination threshold is considered. In the scenario of the registration error *Q* = 5, the target size *D* = 15 m and JNR = 7.5 dB, Figure 8 shows the target discrimination results as a function of the discrimination threshold κ, the simulation results are obtained by averaging 10^5^ trials. Obviously, with the increase in the threshold κ, the discrimination probability of true targets gradually increases until it is close to 1, and the misjudgment probability of false targets also increases slowly. Therefore, in order to obtain both a high discrimination probability of true targets and a low misjudgment probability of false targets, it is of significant importance to select the discrimination threshold, which cannot be too high or too low. The simulation results show that the threshold should be larger than one and lower than the number of false targets generated by one jammer minus two. However, the number of false targets is always unknown for the radar, the discrimination threshold can be chosen as a smaller integer greater than one.

## 5. Conclusions

In the distributed multiple-radar system, an active false target discrimination algorithm was proposed with the presence of spatial registration error. The size of registration error was derived theoretically depending on the radar site error. With the uncertainty region RARCs, the ML algorithm was used to estimate the RARC where the target locates in each spatial diversity channel, thereby forming the target echo vector. According to the difference in spatial scattering characteristics between true and false targets, active false targets were discriminated by the correlation between target received signal vectors. Monto Carlo simulations were utilized to evaluate the discrimination performance, and its feasibility has been verified. The main merit of the new discriminator lies in that it works under registration error and can discriminate false targets in the scenario of multiple independent jamming sources. However, this paper considers only the discrimination of aerial targets. There is no fading effect. In the complex multipath or occlusion scenarios, the jamming signal may be de-correlated, leading to the deterioration of the discrimination performance. Moreover, in the case of low JNR, the algorithm still has a certain probability of misjudging false targets, which is left for future research.

## Figures and Tables

**Figure 1 sensors-22-07216-f001:**
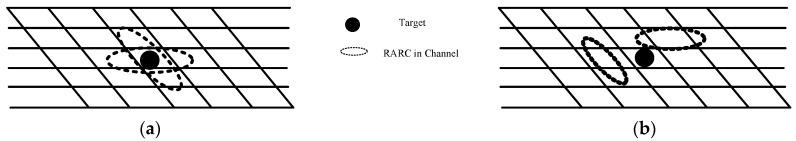
Distribution of RARC locations in the same coordinate system. (**a**) Full registration; (**b**) registration error exists.

**Figure 2 sensors-22-07216-f002:**
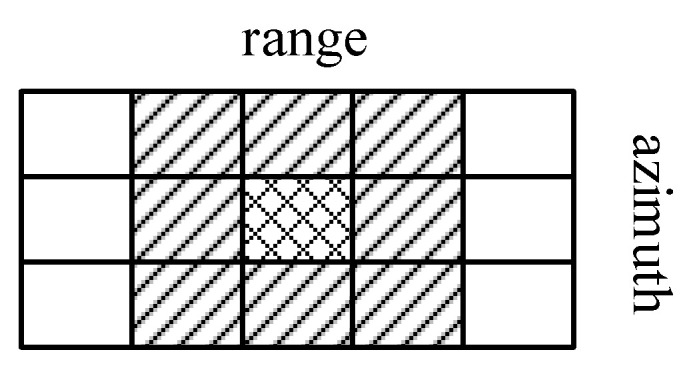
Target uncertainty region in a spatial diversity channel.

**Figure 3 sensors-22-07216-f003:**
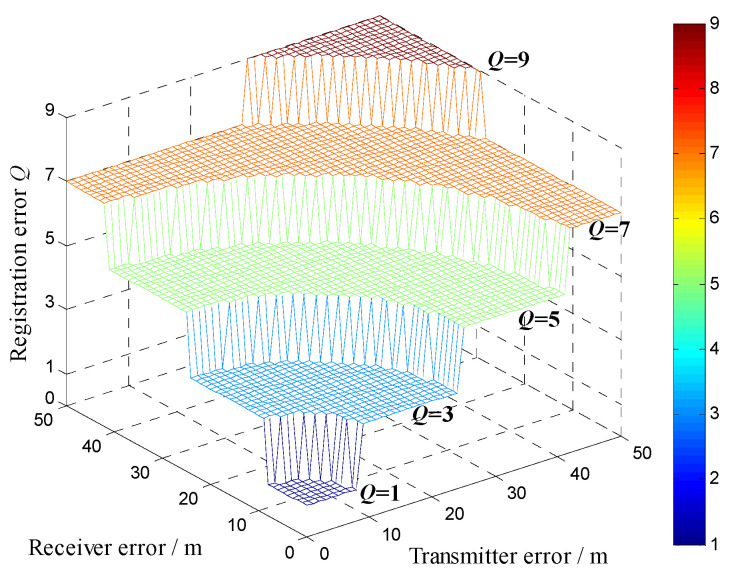
The registration error as a function of the site errors.

**Figure 4 sensors-22-07216-f004:**
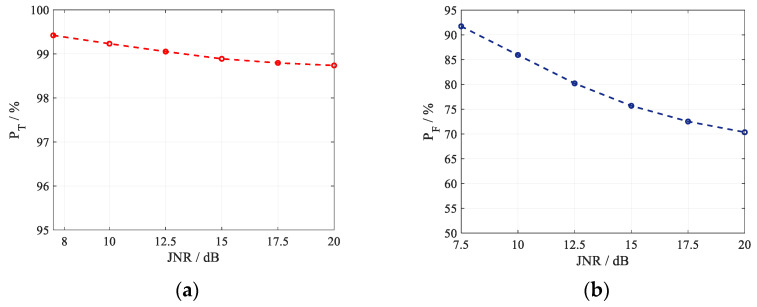
Discrimination performance of the the existing method [27] with *Q* = 3 and *D* = 30 m. (**a**) *P*_T_; (**b**) *P*_F_.

**Figure 5 sensors-22-07216-f005:**
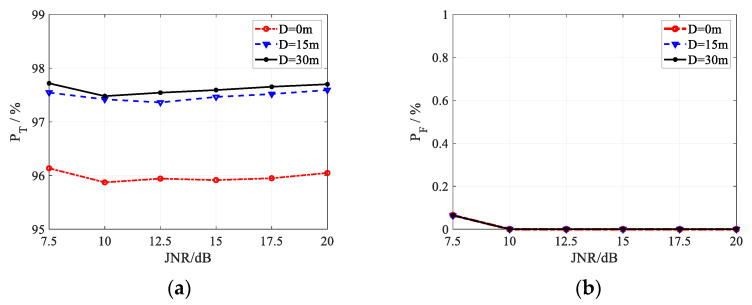
Discrimination performance of the proposed anti-deception jamming method under *Q =* 3. (**a**) *P*_T_; (**b**) *P*_F_.

**Figure 6 sensors-22-07216-f006:**
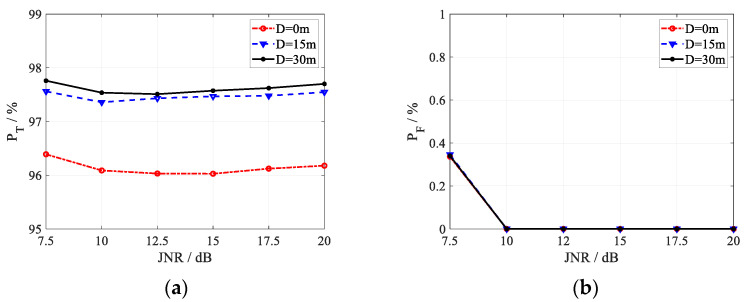
Discrimination performance of the proposed anti-deception jamming method under *Q =* 5. (**a**) *P*_T_; (**b**) *P*_F_.

**Figure 7 sensors-22-07216-f007:**
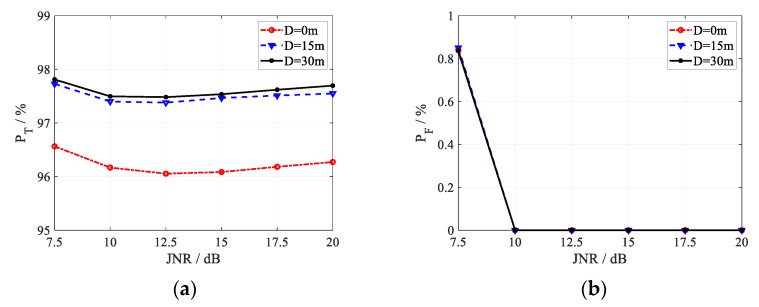
Discrimination performance of the proposed anti-deception jamming method under *Q =* 7. (**a**) *P*_T_; (**b**) *P*_F_.

**Figure 8 sensors-22-07216-f008:**
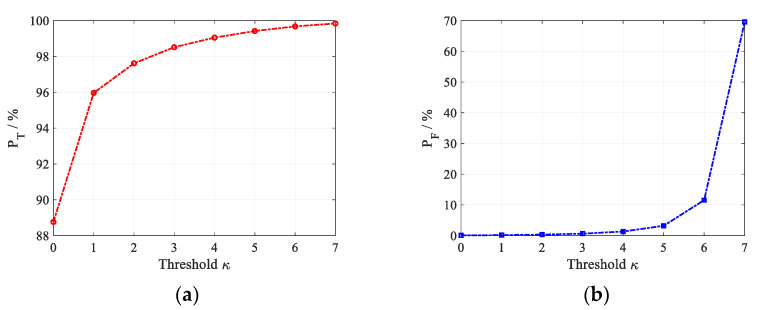
Effect of varying threshold κ on the discrimination performance of the proposed anti-deception jamming method. (**a**) *P*_T_; (**b**) *P*_F_.

**Table 1 sensors-22-07216-t001:** Location coordinates of transmitters, receivers, and targets.

Names	Location Coordinates
Transmitters	[0, 0] m; [300, 0] m
Receivers	[0, 0] m; [±500, 0] m; [±250, 0] m
Targets	[29, 30] km; [30, 29] km; [30, 31] km; [31, 30] km
Jammers	[29, 30] km; [30, 29] km

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
