# Peer review of "Cooperative Anti-Deception Jamming in a Distributed Multiple-Radar System under Registration Errors"

_sensors, 2022, doi:10.3390/s22197216_

Round 1

Reviewer 1 Report

This paper considers the problem of cooperative anti-deception jamming under registration error in distributed multiple-radar system. A theoretical derivation for the registration error is given as a function of the radar site errors. Then, target received signal vectors are estimated from an uncertainty region determined by the registration error by maximum likelihood (ML) algorithm in each channel. With the estimated received signal vectors, target discrimination algorithm is designed based on the difference in target spatial scattering characteristics.

The topic is interesting, and the paper is well written and organized. Before publications some necessary modifications are needed.

1.    In the second paragraph of Section 1, it is said “signal-level fusion algorithms, instead of data-level fusion algorithms have inevitably become the major development tendency, which can achieve better anti-jamming performance.” A sentence or more should be added to explain why signal-level fusion algorithms can achieve better anti-jamming performance.

2.    In Section 1, “full spatial registration” is used to indicate no registration error. But, “complete spatial registration” is used in Section 2. It’s better to have a unified statement.

3.    In Section 3.1.1, the author assumes, the distribution of the received signal of a false target follows the same distribution as the true target, which should be determined by the jammer. Is this required by the algorithm? It is suggested that some explanations should be added for why this assumption is made.

4.    In Section 4, all of the radars are located in a plane, which is a reasonable first assumption, but why are the target and the jammer also located in the same plane. Is it possible to introduce a height coordinate for each of the target and the jammer?

5.    As shown in Figures 4-6, the proposed anti-deception jamming method works even when the true target echoes in distributed stations are highly correlated with the target size being 0. Please provide the reason to reduce possible confusion.

Reviewer 2 Report

The paper presents a method of using the correlation between the signals resulting from a single jammer to detect false targets.  The primary idea is that false targets created by a single jammer will have a correlation which will not be present in signals reflecting off legitimate targets.

Main issues:

11.       No comparison with other techniques for detecting jammers and false signals. This is an old problem; there are several techniques for dealing with jammers that should be discussed here. At minimum, the authors should compare their technique with other jamming detection/rejection methods.

22.       It would seem that multipath fading or shadow fading could also create independent components in the jamming signal which could deter this method.  Why this is not a factor should be mentioned.

33.       Can a jammer use other techniques such as antenna arrays or cooperation with other signals to make their signals uncorrelated?  The consequences of this should be discussed.  Is the additional cost required not practical? The method seems to use MIMO techniques to look for correlations, why can the jammer not use MIMO techniques to remove correlations?

I am not an expert in RADAR design but this would appear to be an easy method that should have been discussed in the prior literature.  The use of MIMO for the legitimate user is explored but if it is available at low cost, why can the jammer not get the benefit at the same cost?

Round 2

Reviewer 2 Report

I think that the authors have answered most of my comments with regards to the first draft of this paper.  The comments on these answers are below.  I only that that some minor additions with respect to these additions are needed.

11.       The comparison with prior art seems to be relevant to this discussion.  It is relevant that the prior art methods do not work as well as the proposed technique for the problems under discussion.

22.       With regards to the problem of multipath propagation and shadow fading de-correlating the signal, the authors are correct in that if only a line-of-sight signal illuminates the targets there will be no fading effects in the received signal but this requires the RADAR system to have already localized the target so that the beam is directly aimed at the target and that there are no scattering objects in the cone of illumination.  This might be true for aerial targets or targets on flat terrain but targets in environments with many reflectors such as urban, mountainous, or hilly environments will result in fading, in particular when the RADAR source is also moving.  It should be noted that the case the authors describe is an important scenario for RADAR so the paper has relevance to practitioners.  It is, however, not universal and this should be noted in the paper.

33.       Your reasoning is sound for how it would be difficult for the jammer to make perfect false targets for tracking. However, if the jammers can create multiple false signals with lack of correlation between some radar receivers, this can increase the computational load of the RADAR system due to the need to separate signals which generate valid tracks from those which create false tracks.  This is in addition to the standard problem of data association for RADAR systems. Obviously, the algorithm in this paper will reject most of the tracks. 

In conclusion, I think that only a minor revision is needed at this point.
